# Quantitative characterization of microstructure and research on spatial variation characteristics of loess of different strata in Luochuan, Shaanxi, China

Yupeng Chang[1]*, Shaoqing Yuan[2]

**1** PowerChina Northwest Engineering Corporation Limited, Xi'an, China, **2** School of Geological Engineering and Geomatics, Chang'an University, Xi'an, China

* 2018126092@chd.edu.cn

**Data Availability Statement:** All relevant data are within the manuscript and its Supporting information files.

## Abstract

The complete sequence of loess strata in Luochuan has become a typical section in loess strata, and is the main focus of research for many scholars studying loess. We were based on the theory of aeolian loess and established a set of quantitative index parameters for loess microstructure through our previous research, such as equivalent diameter, sphericity, morphology ratio, orientation angle Phi, orientation angle Theta, pore Eq-Radius, throat Eq-Radius and throat channelLength. Through the quantitative characterization of various index parameters of the Luochuan loess, we found that the probability density of each index parameter meets a specific distribution well, and in terms of spatial dimension, it shows that as the depth of the strata increases, the average particle size and the mode of pore Eq-Radius, throat Eq-Radius and throat channelLength generally increase, while the mode of particle morphology ratio generally decreases. In addition, loess particles in deeper strata are less prone to vertical sedimentation and tend to deposit gently or horizontally. Most particles in different strata are distributed in a northwest or southwest direction. During the formation period of strata, the main cause for spatial differences is the material carrying force. We conducted a statistical analysis on the correlation between the macroscopic physical properties of loess and its microstructure index parameters. Specifically, we found a positive correlation between loess density and the average particle size and the mode of particle equivalent diameter, Additionally, we found a negative correlation between loess liquid limit and plastic limit, and the mode of particle morphology ratio. Furthermore, there was a negative correlation between permeability coefficient and the mode of pore Eq-Radius, throat Eq-Radius, and throat channelLength.

## Introduction

From the 19th century to the early 20th century, scholars from various countries conducted in-depth research on the causes of loess and proposed many different theories. Lyell believed that the causes of loess were alluvial, floodplain, lake sedimentation, and marine sedimentation. Richthofen proposed the eolian loess theory, and later, other researchers suggested the

**Funding:** The author(s) received no specific funding for this work.

**Competing interests:** The authors have declared that no competing interests exist.

theory of pluvial and slope accumulation. The controversy surrounding the genesis of loess persists to this day. Among them, the eolian loess theory is the most recognized viewpoint among many scholars in various countries at present.

Based on the aeolian Loess theory, many scholars have conducted a series of studies on loess. However, in the early studies, their focus was primarily on the macro physical properties of loess. With the innovation of technology and the deepening of research, scholars have gradually noticed that the microstructure characteristics of loess are the most fundamental factors controlling the structural properties of loess. These characteristics directly influence the macro physical properties and engineering properties of loess. The relevant research findings on the microstructure of loess are crucial for addressing loess engineering challenges. Chinese scholars Haizhi Zhu [1, 2], Zonghu, Zhang [3], and others have pioneered the research of loess microstructure in China. Foreign scholars have also made significant contributions to loess microstructure [4–7] during the same period.

With the continuous development of observation techniques and image processing technology, such as the application of optical microscope, scanning electron microscope (SEM), mercury intrusion method, nuclear magnetic resonance (NMR), and computed tomography (CT), scholars have conducted more specific and detailed research on the microstructure characteristics of loess including particle morphology, pore distribution, particle contact, connection, and arrangement, and have achieved remarkable results [8–12].

In recent years, some domestic scholars have dedicated themselves to exploring the relationship between the microstructure of loess and its macroscopic physical and mechanical properties. Then have conducted research the combines microscopic experiments with macroscopic physical and mechanical experiments, leading to fruitful results. These results could provide a reference basis for multi-scale analysis of loess disasters [13–24]. In his research on the permeability properties of unsaturated loess, Wang Haiman assessed the permeability of densely packed loess and examined the microstructure changes in loess before and after rainfall using SEM and NMR methods. Additionally, a model was developed to estimate the permeability of unsaturated densely packed loess [25]. Modified loess is of great significance in enhancing the engineering properties of loess. The improved loess has demonstrated enhancements in both microstructure and macroscopic strength, making it a research hotspot in recent years [26–28]. Despite these achievements, there are still controversies about the causes and mechanisms of loess geological disasters. At the same time, research results mainly focus on the two-dimensional microstructural characteristics of loess, and few researchers have conducted studies on the three-dimensional microstructural characteristics and quantification of loess.

The three-dimensional microstructure of loess can be obtained through high-resolution scanning of an optical microscope combined with continuous sectioning and three-dimensional image reconstruction technology. NMR is more suitable for detecting pore water in loess. While scanning electron microscopy and CT scanning can provide high-resolution images, they are costly and difficult to access for most researchers. Therefore, considering the economic and technical feasibility, this paper utilizes the optical microscope, which has a higher level of technological maturity, as the primary tool for this study.

The Luochuan loess profile is situated on the Luochuan loess plateau in the central region of the Loess Plateau, with a thickness ranging from 138 to 150 meters. Due to its complete stratigraphic sequence and good exposure, it has become a typical profile in the loess strata and has been favored by many researchers in loess. Therefore, there are many research results on the Luochuan loess [29–33]. However, most of these studies focus on the macroscopic physical properties and engineering characteristics of loess, with limited research on its microstructure. Therefore, we selected eight typical strata of the Luochuan loess, including L1, S1,

L2, S2, L6, L7, L8, and L9 for sampling. Through field and laboratory experiments, we obtained the physical and mechanical parameters of the loess. We conducted research on the three-dimensional microstructural characteristics of the loess in the above strata based using optical microscopy and continuous slicing techniques.

## Materials and experimental method

### Experiment of physical property

In the process of sampling loess from different strata in Luochuan, the destruction of the sections increased the sampling difficulty and risk. To ensure the representativeness of the samples, loess samples were collected from the L1, S1, L2, S2, L6, L7, L8, and L9 strata. Indoor physical and mechanical tests were conducted on loess samples from various strata to determine their physical and mechanical properties. The density, liquid limit, plastic limit, and shear strength of loess samples in each stratum are presented in Table 1. The cohesion (C) and internal friction Angle (φ) were determined through direct shear experiments, while the collapsibility coefficient was obtained through double-line experiments.

### Research method of three-dimensional microstructure of loess

The continuous slicing method based on an optical microscope is used to study the three-dimensional microstructure of loess. The experiment and image reconstruction process (Fig 1) mainly include the following steps. (1) Sample preparation. We cut the loess sample into a cylinder with a diameter of 1 cm and a height of 1.5 cm. Subsequently, we saturated it with a soaking solution in a vacuum environment [34], The soaking solution consists of epoxy resin, acetone, ethylenediamine, and dibutyl phthalate in a volume ratio of 100:50:2:1. After a period of time, the sample completely hardened, and we prepared the hardened sample for modeling. (2) Image acquisition. We utilized a Multiprep device to grind and polish the sample. During this process, we employed a vertical displacement laser monitor and adjusted the polishing duration to control the sample thickness, ensuring a reduction of approximately 2 microns after each polishing cycle. We used the Leica DM6000M with a planar image resolution of 4micron per pixel to observe and capture photos of the polished sample, in order to obtain a two-dimensional image of the sample. Repeat the aforementioned grinding, polishing, observation, and photography process to ultimately acquire a series of 2-micron interval two-dimensional images. (3) Three-dimensional (3D) structure reconstruction. In AVZIO software, we begin by manually aligning images using the Align feature and then utilize the Slices command to automatically align slices. After that, we use the Thresholding tool to determine

**Table 1. Physical parameters of loess samples from different strata in Luochuan.**

| Sampling site | Unit mass $\gamma$(g/cm³) | Liquid limit $L_L$(%) | Plastic limit $P_L$(%) | Permeability coefficient $k$ (cm/s) | Cohesion $C$ (kPa) | Internal friction$\varphi$ (°) | Collapsibility coefficient $\delta_s$ |
|---|---|---|---|---|---|---|---|
| L1 | 1.32 | 16.2 | 11.7 | $1.43\times10^{-4}$ | 63.50 | 23.53 | 0.027 |
| S1 | 1.48 | 28.4 | 21.0 | $4.63\times10^{-5}$ | 78.53 | 15.63 | 0.017 |
| L2 | 1.35 | 30.6 | 20.8 | $4.01\times10^{-5}$ | 54.20 | 20.61 | 0.052 |
| S2 | 1.48 | 31.9 | 21.6 | $4.34\times10^{-5}$ | 101.75 | 18.36 | 0.027 |
| L6 | 1.62 | 30.7 | 20.9 | $3.77\times10^{-5}$ | 61.68 | 31.67 | 0.015 |
| L7 | 1.43 | 31.5 | 22.1 | $3.68\times10^{-5}$ | 62.29 | 29.75 | 0.033 |
| L8 | 1.54 | 32.1 | 22.5 | $3.49\times10^{-5}$ | 37.59 | 32.07 | 0.018 |
| L9 | 1.51 | 32.6 | 23.6 | $3.24\times10^{-5}$ | 80.55 | 30.56 | 0.026 |

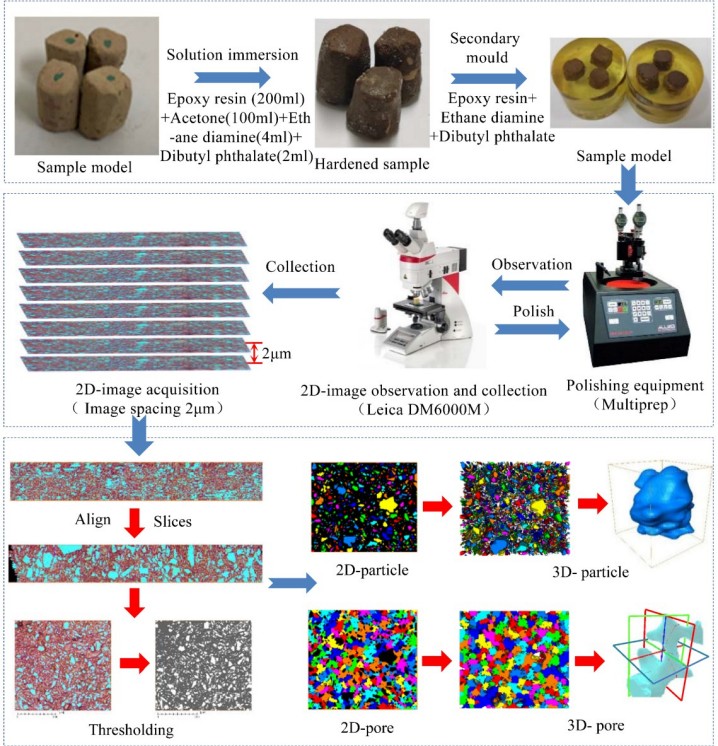

**Fig 1. Experiments and model construction process.** (Reprinted from [12] under a CC BY license, with permission from [Hindawi], original copyright [2020]).

the appropriate threshold for segmenting particles and pores by adjusting the grayscale of the image. After completing threshold segmentation, we extract two-dimensional microstructures such as particles and pores, and then proceed to creating the morphology of three-dimensional particles and pores. (4) Quantitative analysis [35]. Based on the results of the experiment and image reconstruction, the 3D microstructure of loess particles and the three-dimensional pore network were constructed. Various important parameters related to volume, morphology, orientation, etc., were utilized to quantitatively depict the characteristics of loess particles. The volume characteristics could be presented using the equivalent diameter (Eq-D), which can be defined as Eq-D $= \sqrt[3]{\frac{6 \times V}{\pi}}$, where the parameter V represents the volume of a loess particle. The morphological characteristics could be depicted using sphericity and morphology ratio. The sphericity is defined as the ratio of the surface area of spheres with the same volume as loess particles to the surface area of loess particles. The particle morphology ratio is defined as the ratio of the major axis (L) to the minor axis (W) of a particle. The orientation characteristics could be depicted using the parameters Phi and Theta, which represent the dip angle and strike angle of a particle, respectively. The definition of the characterization parameters for loess particle is presented in Fig 2. The pore network is divided into a large number of pores by throats. The pore Eq-Radius is defined as the radius of a sphere which is equal to the pore volume, Throat is defined as a narrow channel connecting adjacent pores. The throat channelLength is defined as the distance between the centers of adjacent pores. Definition of characteristic parameters for loess pore is presented in Fig 3.

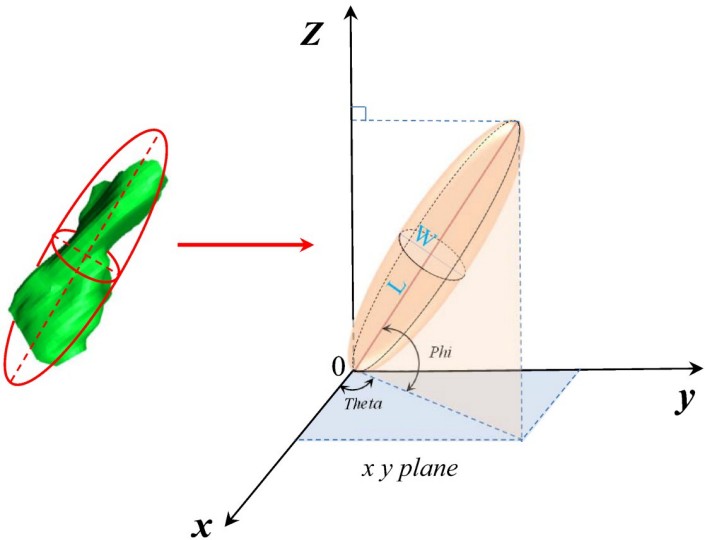

W: Particle width   L: Particle Length

**Fig 2. Schematic diagram of index parameter of loess particle.**

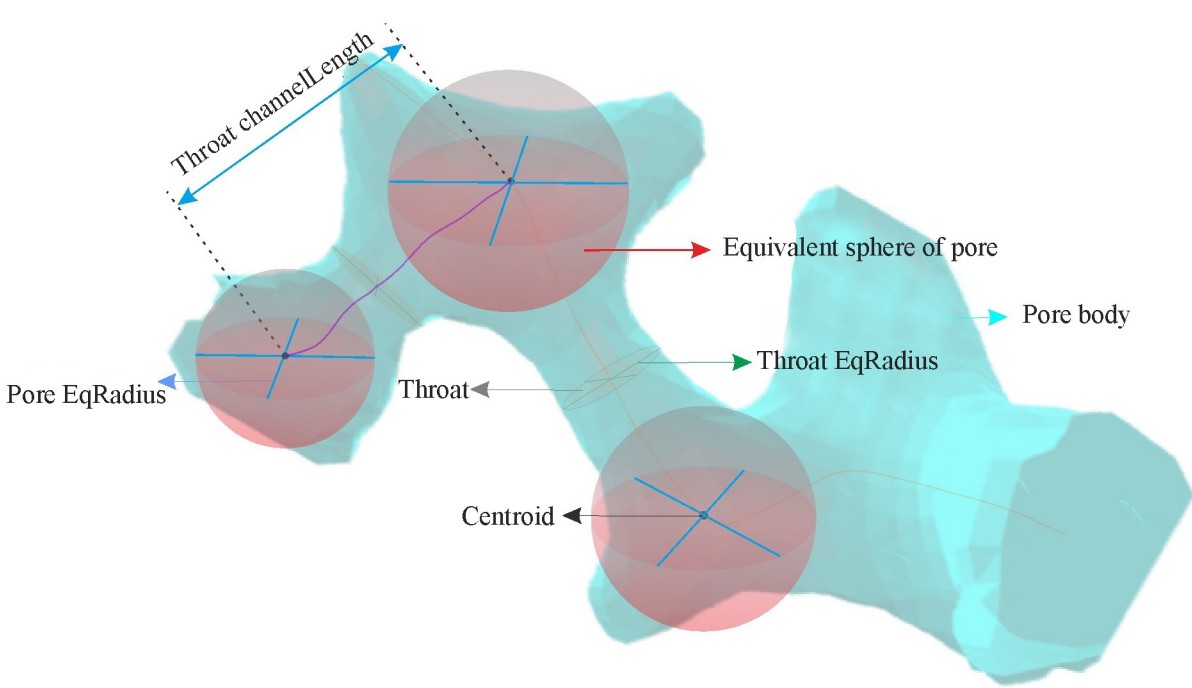

**Fig 3. Schematic diagram of index parameter of loess pore.**

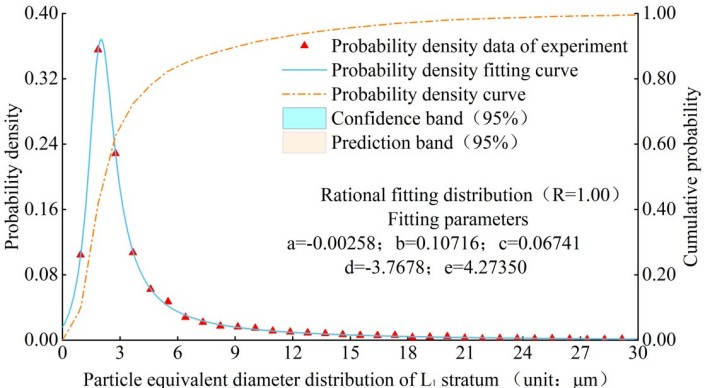

**Fig 4. Probability density distribution of particle equivalent diameter of L1 stratum.**

## Results

### Particle characteristics

**Particle equivalent diameter.**   There are differences in the equivalent diameter of loess in different strata. However, the probability density of the equivalent diameter of loess meets a Rational distribution, Figs 4 and 5 show the experimental results, fitting curve, and cumulative probability curve of the particle equivalent diameter probability density distribution in the L1 stratum and various strata respectively. The probability density function is as follows:

$$f(x) = \frac{(a\,x^2 + b\,x + c)}{(x^2 + d\,x + e)} \tag{1}$$

Where x is the particle equivalent diameter, and a, b, c, d, e are the fitting parameters. According to the fitting situation, and the R value of goodness of fit can exceed 0.95.

Through the analysis and statistics of the modes of the Eq-D and average particle sizes of loess in different strata, it is evident that there is a significant difference in particle size with the changes in loess strata depth. The modes of Eq-D and the average particle sizes of loess in L1,

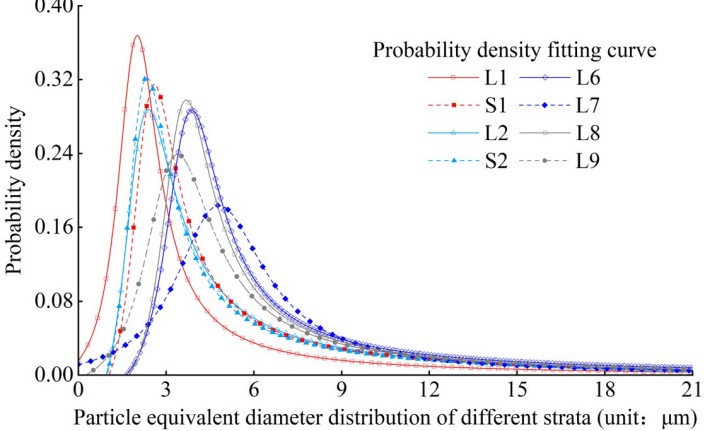

**Fig 5. Comparison of the distribution curves of loess particle equivalent diameter from different strata.**

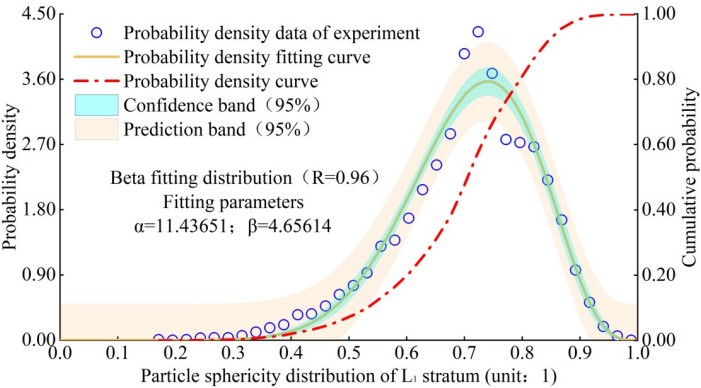

**Fig 6. Probability density distribution of particle sphericity of L1 stratum.**

S1, L2, and S2 strata are significantly smaller than those in L6, L7, L8, and L9 strata. This suggests that the particle size of deeper loess strata is generally larger than that of shallow loess strata.

**Particle morphology.** According to the distribution characteristics of particle sphericity probability density, the sphericity experiment data of loess particles in L1-L9 strata of Luochuan can be well fitted using the Beta distribution (Eq 2). Figs 6 and 7 show the distribution of particle sphericity probability density and fitting curve in the L1 stratum and various strata. The probability density fitting function is as follows:

$$f(x; a,\ b) = \frac{1}{B(\alpha,\ \beta)} x^{\alpha-1}(1-x)^{1-\beta} \tag{2}$$

Where x is the particle sphericity, and $\alpha$ and $\beta$ are fitting parameters. According to the fitting situation of the experimental results, the R value of goodness of fit can exceed 0.95. Comparing the fitting distribution curves in Fig 7, it can be concluded that the overall particle sphericity shows the characteristics of gradual decrease with the increase of strata depth.

The particle sphericity of various strata is distributed between 0.2 and 1.0, but mainly concentrated between 0.6 and 0.85. Except for the L6 and L7 strata, the quantity percentage of

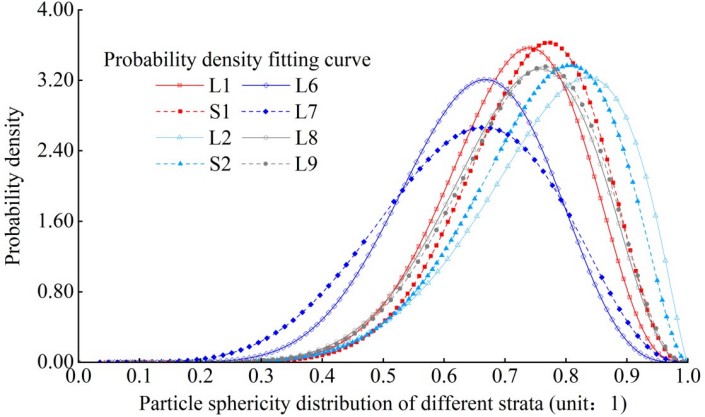

**Fig 7. Comparison of the distribution curves of particle sphericity from different strata.**

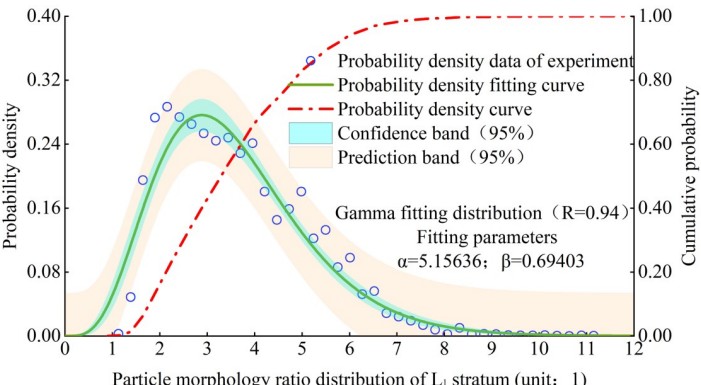

**Fig 8. Probability density distribution of particle morphological ratio of L1 stratum.**

particles with a particle sphericity greater than 0.7 in loess particles in other strata exceeds 50% of the total. According to the classification of particle sphericity characteristics [35], the loess particles could be mainly classified as thin-slice, long-strip, sub-prismatic granule, multi-angled granule, sub-globularity, and globularity, respectively. L6 and L7 strata are mainly composed of multi-angled granule and sub-globularity particles, while remaining strata are primarily composed of sub-globularity particles. Through the analysis and statistics of the modes of particle sphericity in various strata, it was observed that the particle sphericity of L1, S1, L2, and S2 strata gradually increases with the increase of loess strata depth, while the particle sphericity of L6, L7, L8, and L9 strata gradually decrease with the increase of loess strata depth.

Figs 8 and 9 show the distribution of particle morphology ratio probability density and fitting curve in the L1 stratum and various strata. Its distribution meets the Gamma distribution (Eq 3), and the R value of goodness of fit can exceed 0.90. The probability density function is as follows:

$$f(x;\ a,\ b) = \frac{b^a}{\Gamma(a)} x^{a-1} e^{-bx} \tag{3}$$

Where x is the particle morphology ratio, and a and b are the fitting parameters.

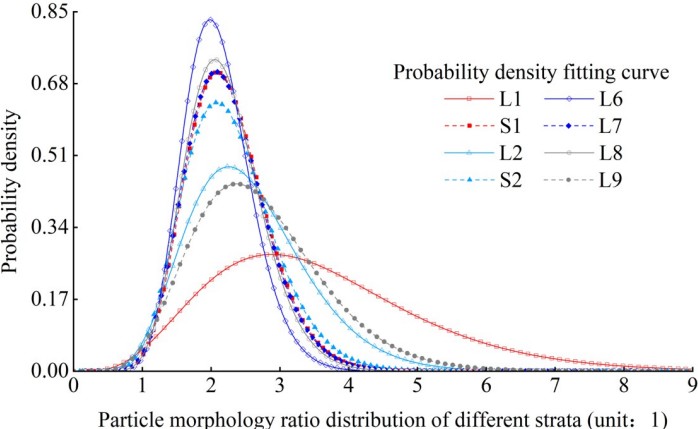

**Fig 9. Comparison of the distribution curves of particle morphological ratio from different strata.**

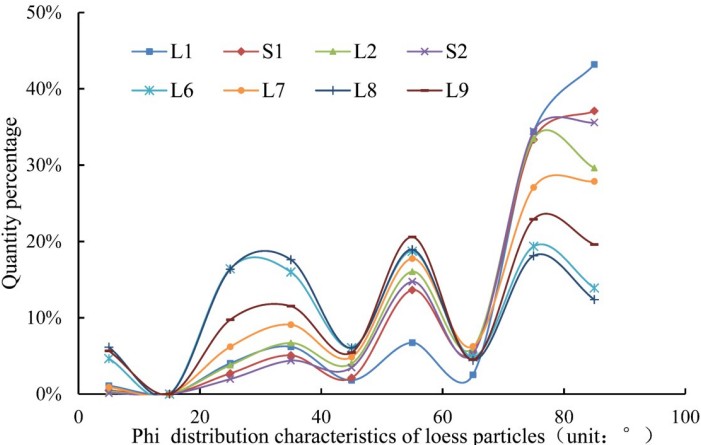

**Fig 10. Particle Phi angle distribution.**

It can be seen that the mode of particle morphology ratio generally shows a de-creasing trend with the increase of loess strata depth, as depicted in Fig 9.

**Particle orientation.** Fig 10 shows the percentage distribution curve of particle Phi angle in L1-L9 strata of Luochuan. The changes in particle Phi angles in L1-L9 strata show similar in-crease and decrease characteristics. When comparing the quantity percentage of Phi angles ranging from 0 to 50 degrees, it is evident that the deeper strata consistently exhibit a higher percentage than the shallow strata. This suggests that particles in the deeper layers of loess are less likely to be deposited vertically and tend to settle more gently or horizontally. Fig 11 shows the percentage distribution curve of particle Theta angle in L1-L9 strata of Luochuan loess. The trend of variation in the Theta angle of particles in strata L1-L9 is consistent. Except for the L6 stratum, the dominant angles of the other strata are concentrated between 120° and 160°. When the positive direction of the X-axis is due south, most of the particles in each stratum are distributed in the northwest or southwest direction. The particle orientation

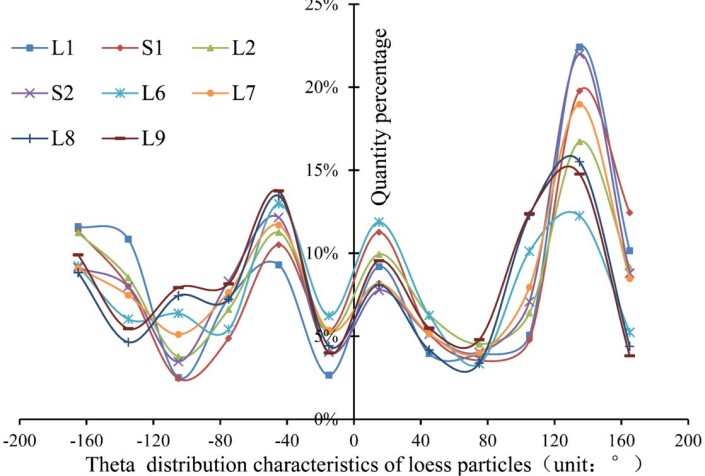

**Fig 11. Particle Theta angle distribution.**

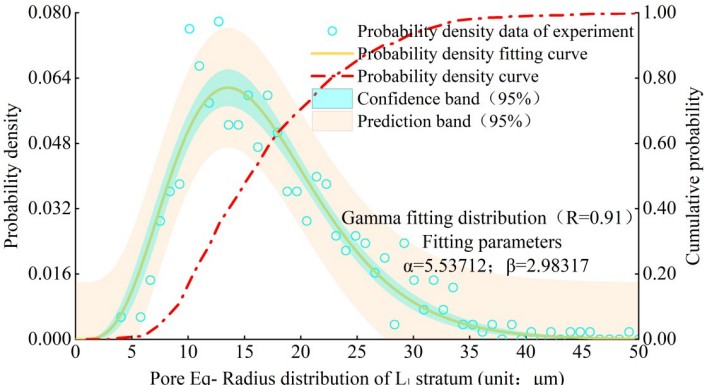

**Fig 12. Probability density distribution of pore Eq-Radius of L1 stratum.**

characteristics of loess strata in different ages are basically similar within the same sedimentary site. The orientation characteristics of particles are primarily related to the plane position of the Loess Plateau and are predominantly governed by material carrying distance and force.

## Pore and throat characteristics

According to the experimental results, the pores were extracted from the image using AVZIO software, and the three-dimensional pore model (Fig 3) was established. The quantitative analysis and description of loess pores in different strata in Luochuan are achieved using three quantitative index parameters: pore Eq-Radius, throat Eq-Radius and throat channelLength.

Figs 12 and 13 show the probability density distribution of the pore Eq-Radius in the L1 stratum and various strata. From the distribution characteristics, it can be seen that the fitting function of the probability density distribution of the pore Eq-Radius also meets the gamma distribution, and the R value of goodness of fitting can exceed 0.90.

According to the pore size classification standard proposed by Lei Xiangyi [36], the quantity percentage of micropores and small pores in strata L1-L9 is extremely low. The quantity percentage of meso-pores are 52.1%, 49.9%, 35.0%, 40.0%, 32.5%, 32.4%, 40.0%, and 32.5%, respectively. The quantity percentage of macro-pores are 47.5%, 50.0%, 65.0%, 60.0%, 67.5%,

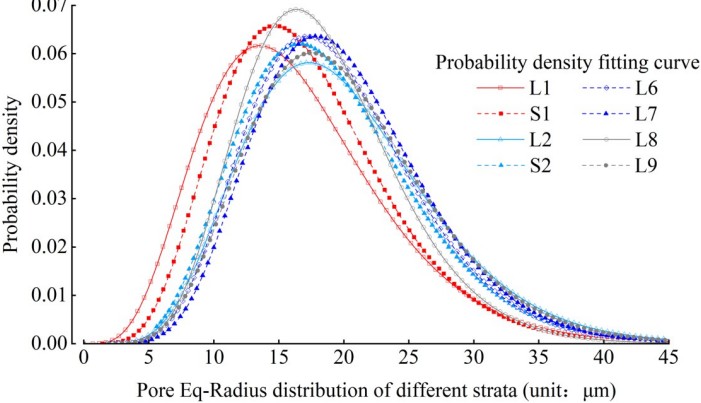

**Fig 13. Comparison of the distribution curves of loess pore Eq-Radius diameter from different strata.**

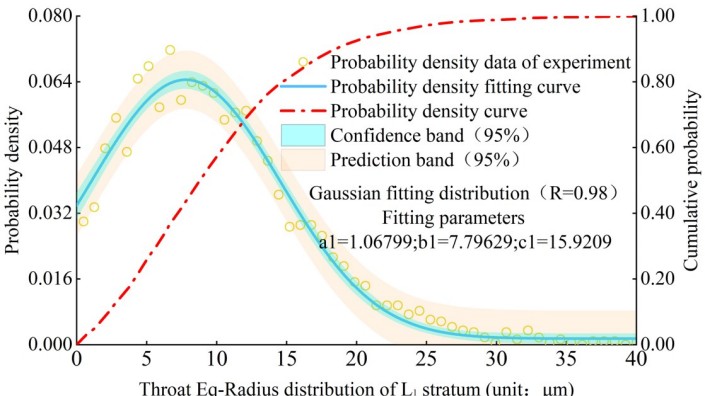

**Fig 14. Probability density distribution of throat Eq-Radius of L1 stratum.**

60.0%, and 67.5%, respectively. It shows that there are almost no micro-pores and small pores in each stratum of loess, which are mainly meso-pores and macro-pores, and most of them are macro-pores; On the whole, the quantity percentage of meso-pores decreases with the increase of loess strata depth, on the contrary, the quantity percentage of macro-pores increases with the increase of loess strata depth. It can be concluded that the mode of pore Eq-Radius generally increases with the increase of loess strata depth through the analysis and statistics of the mode of pore Eq-Radius in each stratum.

Throat size is the main factor determining the permeability and pore connectivity of loess. Based on the results of microstructure experiments and data analysis, the probability density distribution of the throat Eq-Radius was obtained. From the distribution characteristics, it can be seen that the probability density distribution fitting function of the throat Eq-Radius meets the Gaussian distribution (Eq 4). The probability density function is as follows:

$$f(x) = a \times e^{-1 \times \left(\frac{x-b}{c}\right)^2} \tag{4}$$

Where x is the throat Eq-Radius, and a, b, and c are the fitting parameters. Figs 14 and 15

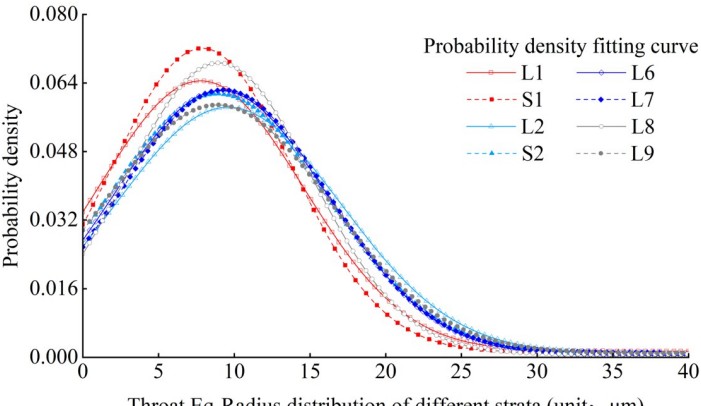

**Fig 15. Comparison of the distribution curves of loess throat Eq-Radius diameter from different strata.**

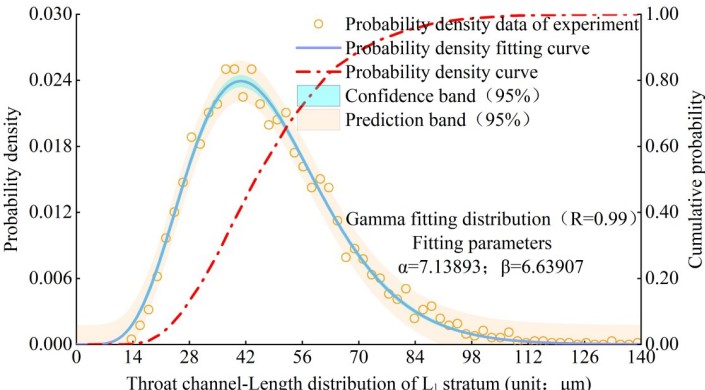

**Fig 16. Probability density distribution of throat channelLength of L1 stratum.**

show the probability density distribution of the throat Eq-Radius in L1 stratum and various strata, and R value of goodness of fitting can be close to 1.

The throat Eq-Radius in each stratum is primarily distributed between 0 micron and 35microns, and the mode of throat Eq-Radius is mainly distributed between 7 microns and 10microns. The number of throats with a radius between 0 micron and 10microns in each stratum can reach approximately 50%. According to statistical analysis, the modes of throat Eq-Radius in L1-L9 strata are 7.8microns, 7.5microns, 9.5microns, 8.7microns, 9.0microns, 9.1microns, 9.1 microns, and 9.0 microns, respectively. The mode of throat Eq-Radius shows a gradually increasing characteristic with the increase of loess strata depth.

Based on the experimental results, the probability density distribution characteristics of the throat channelLength in L1-L9 strata of Luochuan loess were obtained. Figs 16 and 17 show the probability density distribution and fitting curves of the throat channelLength, respectively, in the L1 stratum and various strata. It can be seen that the probability density distribution fitting function of the throat channelLength meets the Gamma distribution well.

According to statistical analysis, the modes of throat channelLength in L1-L9 strata are 41microns, 40.5microns, 44.5microns, 43.5microns, 44.5microns, 45.5microns, 41.0microns,

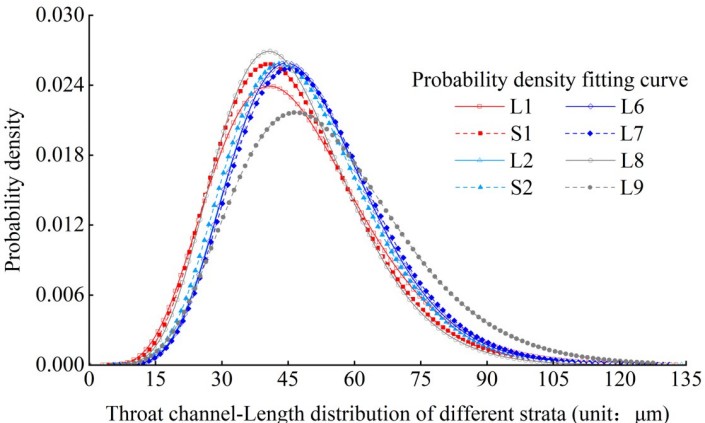

**Fig 17. Comparison of the distribution curves of loess throat channelLength diameter from different strata.**

and 46.5 microns, respectively. It shows the mode of the throat channelLength in L1-L9 strata of Luochuan loess has an gradually increasing characteristic with the increase of loess strata depth.

By comparing the distribution of pore Eq-Radius and throat Eq-Radius in different strata, it can be observed that the sizes of pores and throats in deeper loess strata are larger. It is inferred that the reason for this phenomenon is related to the particle sizes of loess in different strata. The loess in deeper strata has larger particles, smaller sphericity, and is prone to the formation of macropores, mostly in the form of particle aerial structures [36]. The loess in shallower strata has smaller particles, larger sphericity, and more surface-to-surface contact between particles, making it prone to the formation of mesopores and small pore.

## Discussion

### Analysis of differences of loess strata in Luochuan

During the lengthy deposition process of loess, the loess particle sizes in different strata exhibit certain variations. Based on the micro-scale study of loess particles in the above strata, it can be observed that the loess particles tend to become larger with the increase of loess strata depth as shown in Fig 5. The change of loess particle sizes in different strata is not only associated with the particle size of the original loess parent material and the material handling force, but also with the change of climate conditions and weathering during the formation of strata to some extent. The primary material source of the Loess Plateau comes from the desert of Inner Mongolia next to Mongolia in the northwest. Based on this, it can be inferred that the material source of each stratum is essentially the same. Therefore, the material carrying force has become the main reason for the difference of loess particle size within the aforementioned strata. The particle sizes of L6, L7, L8, and L9 strata are large, suggesting that there was strong carrying wind and dry and cold climate during that time, but L8 and L9 strata underwent significant physical weathering, resulting in slightly smaller particle sizes compared to L6 and L7 strata. The particle sizes of L1, S1, L2, and S2 strata are small, suggesting that the carrying wind force was relatively weakened during that time and suffered different degrees of weathering during carrying process. It can be inferred that most of the loess materials forming L6, L7, L8, and L9 strata, such as silts, were suspended in the air during the carrying of strong wind force, resulting in relatively few instances of abrasion between them. At the same time, medium sand, fine sand, and other materials were swiftly transported to the vicinity of loess accumulation remotely by leaps and bounds under the carrying of strong wind force. The loess materials forming L1, S1, L2, and S2 strata, such as silts, were suspended close to the surface in the air and carried by the carrying of weak wind force, causing constant abrasion of the materials. At the same time, some medium sands and fine sands carried from a long distance and near the loess accumulation were carried twice or repeatedly under the carrying of weak wind force, and finally loess particles with high roundness and poor sorting were formed.

We have conducted preliminary explorations and inferences on the genesis of the Luochuan loess based on microstructure quantitative index parameters. We have clarified that the material sources of the loess of various strata in Luochuan were basically the same during the formation period. The material carrying force was the main cause of the differences in the properties of the various strata. This research result is essentially consistent with the overall environment of the formation of the Loess Plateau. It is also consistent with the current mainstream research results [37–40]. However, we need to conduct more research on the Luochuan loess from multiple perspectives to confirm our hypothesis. In the next research, we will try to combine stratum magnetic susceptibility with microscopic structural parameters to conduct further research, revealing the paleoclimatic environment during the formation period of the

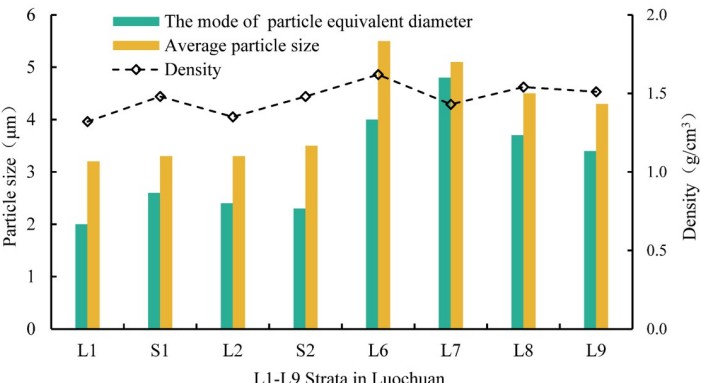

**Fig 18. Correlation between the density and average particle diameter and mode of particle equivalent diameter.**

loess of various strata in Luochuan. In the study, we found that the quantity percentage of micro-pores and small pores in the loess of various strata in Luochuan is extremely low. This feature may be influenced by the experimental method used, and a large number of pores less than 2 microns are ignored due to continuous slicing.

## The correlation between physical properties and microstucture index parameters of loess

A highlight of our research results is that we revealed the intrinsic relationship between loess microstructure and physical properties through quantitative characterization of microstructural indicators. This approach differs from previous methods that relied loess microstructure to investigate physical properties [41–45], offering new research ideas for subsequent scholars. In our research results, the density of loess is significantly correlated with the average particle size and the mode of particle equivalent diameter overall. Additionally, the liquid limit and plastic limit are significantly correlated with the mode of morphology ratio overall. The permeability coefficient is significantly correlated with the mode of pore Eq-Radius and throat Eq-Radius on the whole. However, its shear strength parameters and collapsibility coefficient do not demonstrate a correlation with the microstructural index parameters. This lack of correlation may be influenced by the limited sample data available.

The physical properties of loess are closely associated with the characteristics of loess particles, pores, and throats. Among these factors, the density, average particle diameter, and the mode of particle equivalent diameter of L1-L9 strata increase overall with the increase of strata depth as depicted in Fig 18, indicating that the density of loess shows a positive correlation with the average particle size and the mode of particle equivalent diameter. The increase in density of Luochuan loess is related to the increase in particle size to some extent.

Fig 19 shows the correlation between the plastic limit, liquid limit and the mode of particle sphericity and particle morphology ratio. The plastic limit and liquid limit exhibit a significant negative correlation with the mode of morphology ratio, while the plastic limit and liquid limit show a weak negative correlation with the mode of particle sphericity. With the increase of loess strata depth, the density and average particle size of loess also increase, and the particle morphology ratio of loess overall decreases and the quantity percentage of thin-slice and long-strip particles decreases, and the quantity percentage of sub-prismatic granule, multi-angled granule, and sub-globularity particles relatively increases. This leads to an increase in the specific surface area of loess, subsequently increasing the liquid limit and plastic limit.

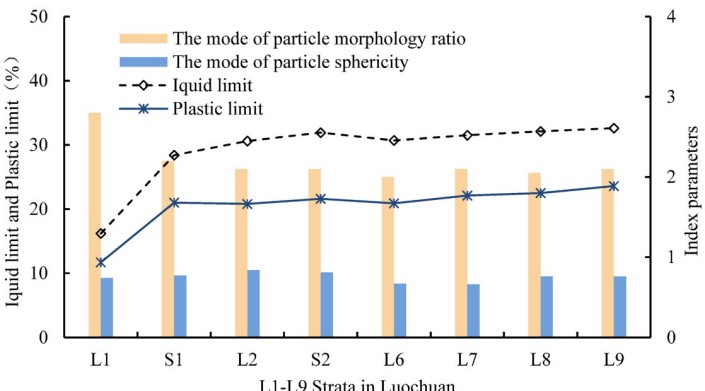

**Fig 19. Correlation between liquid -plastic limit and the mode of particle sphericity and particle morphology ratio.**

Fig 20 illustrates the correlation between permeability coefficient and the mode of pore Eq-Radius and throat Eq-Radius. The mode of pore Eq-Radius and throat Eq-Radius increase with the increase of strata depth, while the permeability coefficient decreases significantly with the increase of strata depth, indicating a negative correlation between the permeability coefficient and their mode. Fig 21 illustrates the correlation between the permeability coefficient and the mode of throat channelLength, and it shows a negative correlation between the permeability coefficient and the mode of throat channelLength. Based on this, it can be inferred that the permeability behavior of Luochuan loess primarily depends on the quantity variations of pores and throats unit volume, and there is little correlation with the size of pores and throats.

## Conclusions

The probability density of the parameters of loess particles and pores microstructure meet a specific functional distribution well.

The overall characteristics of the loess in Luochuan show that with the increase of loess strata depth, the size of particle, pore and throat, and throat channelLength gradually increase, and the particle morphology ratio gradually decreases. L1, S1, L2, and S2 strata were formed

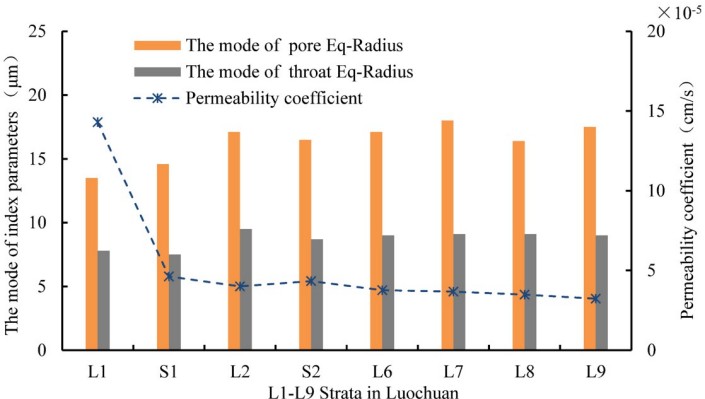

**Fig 20. Correlation between permeability coefficient and the mode of pore Eq-Radius and throat Eq-Radius.**

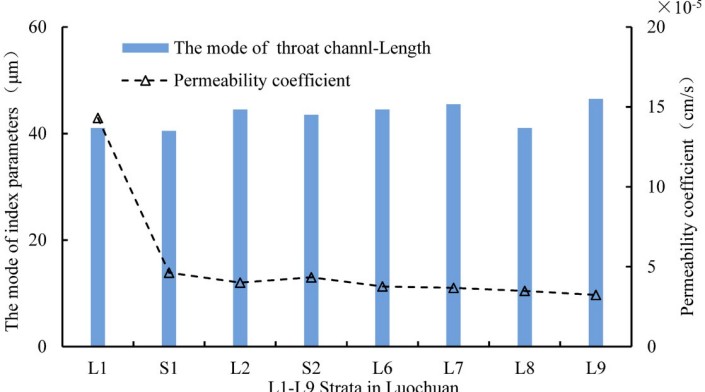

**Fig 21. Correlation between permeability coefficient and the mode of throat channelLength.**

during the weak wind period in the ancient climate, while the L6, L7, L8, and L9 strata were formed during the strong wind period in the ancient climate. The variation trends of particle Phi angle and Theta angle are basically the same, and loess particles in deeper strata are not easy to deposit vertically and more tend to deposit gently or horizontally. Most particles in different strata are distributed in a northwest or southwest direction. The pores distribution of loess in various strata is mainly dominated by mesopores and macropores, but in deeper strata, macropores account for a larger proportion and are mostly single particle stacked pore structure.

The physical properties of loess are closely related to its microstructure parameters. The density of loess has a positive correlation with the average particle size and the mode of particle equivalent diameter. The plastic limit and liquid limit have a negative correlation with the mode of morphology ratio. The permeability coefficient has a negative correlation with the mode of pore Eq-Radius, throat Eq-Radius and throat channelLength, inferring that the permeability behavior of Luochuan loess primarily depends on the quantity rather than size variations within unit volume for both pores and throats.

## Supporting information

**S1 Data set.**
(XLSX)

## Acknowledgments

Thank you to the friends and colleagues who have helped during the experiment, and also to three-dimensional microstructure research laboratory of loess at Chang'an University.

## Author Contributions

**Data curation:** Yupeng Chang, Shaoqing Yuan.

**Formal analysis:** Yupeng Chang, Shaoqing Yuan.

**Investigation:** Yupeng Chang, Shaoqing Yuan.

**Methodology:** Yupeng Chang.

**Project administration:** Shaoqing Yuan.

**Software:** Yupeng Chang, Shaoqing Yuan.

**Supervision:** Shaoqing Yuan.

**Validation:** Shaoqing Yuan.

**Writing – original draft:** Yupeng Chang.

**Writing – review & editing:** Yupeng Chang.

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
