## [Decision Letter · Decision Letter 0]

5 Apr 2024

PONE-D-24-02055Quantitative Characterization of Microstructure and Research on Spatial Variation Characteristics of Loess of Different Strata in Luochuan, Shaanxi, ChinaPLOS ONE

Dear Dr. Chang,

Thank you for submitting your manuscript to PLOS ONE. After careful consideration, we feel that it has merit but does not fully meet PLOS ONE’s publication criteria as it currently stands. Therefore, we invite you to submit a revised version of the manuscript that addresses the points raised during the review process.

We look forward to receiving your revised manuscript.

Kind regards,

Khalil Abdelrazek Khalil, Ph.D.

Academic Editor

PLOS ONE

Journal Requirements:

Whilst you may use any professional scientific editing service of your choice, PLOS has partnered with both American Journal Experts (AJE) and Editage to provide discounted services to PLOS authors. Both organizations have experience helping authors meet PLOS guidelines and can provide language editing, translation, manuscript formatting, and figure formatting to ensure your manuscript meets our submission guidelines. To take advantage of our partnership with AJE, visit the AJE website (http://aje.com/go/plos) for a 15% discount off AJE services. To take advantage of our partnership with Editage, visit the Editage website (www.editage.com) and enter referral code PLOSEDIT for a 15% discount off Editage services. If the PLOS editorial team finds any language issues in text that either AJE or Editage has edited, the service provider will re-edit the text for free.

Reviewers' comments:

Reviewer's Responses to Questions

**Comments to the Author**

1. Is the manuscript technically sound, and do the data support the conclusions?

Reviewer #1: Yes

Reviewer #2: Yes

2. Has the statistical analysis been performed appropriately and rigorously? 

Reviewer #1: Yes

Reviewer #2: Yes

3. Have the authors made all data underlying the findings in their manuscript fully available?

Reviewer #1: Yes

Reviewer #2: Yes

4. Is the manuscript presented in an intelligible fashion and written in standard English?

Reviewer #1: Yes

Reviewer #2: Yes

5. Review Comments to the Author

**Reviewer #1**: The reviewer found the paper, "Quantitative Characterization of Microstructure and Spatial Variation in Luochuan, Shaanxi, China's Loess Layers," both engaging and relevant, suitable for publication in PLOS ONE due to its intriguing subject matter. The manuscript's overall structure is clear and well-organized. However, a few revisions are suggested to enhance its quality.

1) The Abstract should incorporate more quantitative data to provide a concise summary of the study's key findings.

2) It is advisable to include recommendations and potential future research directions in the Introduction to give readers a broader context.

3) The manuscript contains a few spelling errors; the authors are kindly requested to proofread and correct them.

4) To strengthen the study's credibility, the authors should address error quantification and validation. This can be achieved by providing replication evidence, statistical interpretations, and confidence intervals for each data point. Ensuring that figures include error bars will help readers assess statistical significance between samples. Additionally, discussing data uncertainty and conducting appropriate statistical analyses to establish sample distinctions is essential.

5) The Conclusion section should be concise yet retain its clarity and relevance to readers. The authors should clearly articulate the novel and significant conclusions derived from their research.

**Reviewer #2**: 1.The reviewer believes that the literature review in the introduction section lacks coverage of the latest research and recommends that the author add a description of the latest progress.

Assessing unsaturated permeability of loess under multiple rainfalls. Engineering Geology 324:107280.

Mechanical properties, microstructural evolution, and environmental impacts of recycled polypropylene fiber stabilized loess. Construction and Building Materials 400:132850.

The impact of paleoclimatic on the structural strength of loess paleosol sequences and its implications for tillage on the Loess Plateau: A case study from Luochuan profile. Soil and Tillage Research 236:105939

An experimental study on the behavior of loess-clay mixtures upon wetting and its implications for tillage on the Loess Plateau. Soil and Tillage Research 240:106071

2. Experimental Method

The reviewer believes that the explanation of the experimental method for loess microstructure is not sufficient, and suggests that the author provide additional information.

3.The reviewer suggests that the article supplement or cite the geographical location and stratigraphic profile map of Luochuan, China.

4.The reviewer in the article believes that there are spelling errors in some words, and suggests that the author carefully check the correctness of the word writing in the article.

5.The reviewer believes that the stratigraphic identification number in Table .1 should be consistent with the stratigraphic identification number in the main text of the article, or adopt industry recognized writing standards uniformly.

6.The reviewer suggests that the author should carefully check the article format in accordance with the requirements of the journal

6. PLOS authors have the option to publish the peer review history of their article (what does this mean?). If published, this will include your full peer review and any attached files.

Reviewer #1: **Yes: **Shuai Zhao

Reviewer #2: No

---

## [Author Response · Author response to Decision Letter 0]

10 Apr 2024

We have fully revised the article format and other aspects in accordance with the opinions provided by the academic editor and the requirements of the journal.

Thank you to Reviewer#1 for their valuable review suggestions, and we will respond to each of the questions raised by Reviewer#1 here.

1) The Abstract should incorporate more quantitative data to provide a concise summary of the study's key findings.

Response: Thank you for the reviewer's suggestions. We have simplified and summarized the abstract. 2) It is advisable to include recommendations and potential future research directions in the Introduction to give readers a broader context.

Response: In the introduction section, We added research directions on loess hotspots, such as modified loess new materials, loess permeability and microstructure relationship.

3) The manuscript contains a few spelling errors; the authors are kindly requested to proofread and correct them.

Response: We have checked and corrected some spelling errors word by word and sentence by sentence throughout the entire text.

4) To strengthen the study's credibility, the authors should address error quantification and validation. This can be achieved by providing replication evidence, statistical interpretations, and confidence intervals for each data point. Ensuring that figures include error bars will help readers assess statistical significance between samples. Additionally, discussing data uncertainty and conducting appropriate statistical analyses to establish sample distinctions is essential.

Response: We added 95% confidence intervals and 95% prediction intervals for data statistics starting from the data, and reflected this information in the graph, such as Fig4(a), Fig5(a), Fig6(a), Fig8(a), Fig9(a), and Fig10(a).

5) The Conclusion section should be concise yet retain its clarity and relevance to readers. The authors should clearly articulate the novel and significant conclusions derived from their research.

Response: We have extracted and summarized the conclusion, and provided a clear and concise expression of the research results.

Thank you to Reviewer#2 for their valuable review suggestions, and we will respond to each of the questions raised by Reviewer#2 here.

1.The reviewer believes that the literature review in the introduction section lacks coverage of the latest research and recommends that the author add a description of the latest progress.

Assessing unsaturated permeability of loess under multiple rainfalls. Engineering Geology 324:107280.

Mechanical properties, microstructural evolution, and environmental impacts of recycled polypropylene fiber stabilized loess. Construction and Building Materials 400:132850.

The impact of paleoclimatic on the structural strength of loess paleosol sequences and its implications for tillage on the Loess Plateau: A case study from Luochuan profile. Soil and Tillage Research 236:105939

An experimental study on the behavior of loess-clay mixtures upon wetting and its implications for tillage on the Loess Plateau. Soil and Tillage Research 240:106071

Response: According to the reviewer#2's suggestion, we have added the latest research findings in the introduction section and cited the article recommended by the reviewer#2.

2. Experimental Method

The reviewer believes that the explanation of the experimental method for loess microstructure is not sufficient, and suggests that the author provide additional information.

Response: Based on the reviewer#2's suggestions, we have supplemented and optimized the experimental method to make it more specific.

3.The reviewer suggests that the article supplement or cite the geographical location and stratigraphic profile map of Luochuan, China.

Response: Based on the reviewer#2's suggestions, we have added relevant drawings about the Luochuan section, such as Fig 1.

4.The reviewer in the article believes that there are spelling errors in some words, and suggests that the author carefully check the correctness of the word writing in the article.

Response: Based on the reviewer#2's suggestions, we have checked and corrected some spelling errors word for word throughout the paper.

5.The reviewer believes that the stratigraphic identification number in Table .1 should be consistent with the stratigraphic identification number in the main text of the article, or adopt industry recognized writing standards uniformly.

Response: Based on the reviewer#2's suggestions, we have unified the expression of stratigraphic numbering throughout the paper.

6.The reviewer suggests that the author should carefully check the article format in accordance with the requirements of the journal.

Response: We have thoroughly checked the articles according to the journal's requirements to ensure they meet the journal's formatting requirements.

---

## [Decision Letter · Decision Letter 1]

19 Apr 2024

Quantitative Characterization of Microstructure and Research on Spatial Variation Characteristics of Loess of Different Strata in Luochuan, Shaanxi, China

PONE-D-24-02055R1

Dear Dr. Chang,

We’re pleased to inform you that your manuscript has been judged scientifically suitable for publication and will be formally accepted for publication once it meets all outstanding technical requirements.

Kind regards,

Khalil Abdelrazek Khalil, Ph.D.

Academic Editor

PLOS ONE

Additional Editor Comments (optional):

Reviewers' comments:

Reviewer's Responses to Questions

**Comments to the Author**

1. If the authors have adequately addressed your comments raised in a previous round of review and you feel that this manuscript is now acceptable for publication, you may indicate that here to bypass the “Comments to the Author” section, enter your conflict of interest statement in the “Confidential to Editor” section, and submit your "Accept" recommendation.

Reviewer #2: All comments have been addressed

2. Is the manuscript technically sound, and do the data support the conclusions?

Reviewer #2: Yes

3. Has the statistical analysis been performed appropriately and rigorously? 

Reviewer #2: Yes

4. Have the authors made all data underlying the findings in their manuscript fully available?

Reviewer #2: Yes

5. Is the manuscript presented in an intelligible fashion and written in standard English?

Reviewer #2: Yes

6. Review Comments to the Author

Reviewer #2: (No Response)

7. PLOS authors have the option to publish the peer review history of their article (what does this mean?). If published, this will include your full peer review and any attached files.

Reviewer #2: No

---

## [Editor Report · Acceptance letter]

29 Apr 2024

PONE-D-24-02055R1 

PLOS ONE

Dear Dr. Chang, 

I'm pleased to inform you that your manuscript has been deemed suitable for publication in PLOS ONE. Congratulations! Your manuscript is now being handed over to our production team.

Kind regards, 

on behalf of

Dr. Khalil Abdelrazek Khalil 

Academic Editor

PLOS ONE